

# Multiple Equilibrium Configurations in River-Dominated Deltas

Lorenzo Durante[1], Nicoletta Tambroni[1], and Michele Bolla Pittaluga[1]

[1]Department of Civil, Chemical and Environmental Engineering, University of Genova, Genova, Italy.

**Correspondence:** Lorenzo Durante (lorenzo.durante@edu.unige.it)

**Abstract.** The morphological evolution of river deltas is a complex process influenced by both natural forces and human interventions. As hubs of human settlement and economic activity, deltas face unique challenges that necessitate integrated management strategies to balance development with ecological sustainability. This study investigates multiple equilibrium states within deltaic systems, revealing key internal feedback mechanisms between delta branches through a novel theoretical
model tailored to river-dominated delta channels. Focusing on the Po River Delta as a case study, we analyze equilibrium variability and identify potential avulsion sites. These findings provide a valuable framework for predicting future shifts in deltaic morphology and supporting adaptive management strategies.

## 1 Introduction

River deltas are dynamic landforms where fluvial and marine environments merge. The continuous evolution of river deltas is primarily governed by the quantity and the quality of sediment delivered by the river. Sediment load depends on upstream conditions such as natural erosion, land use, damming, and vegetation cover. The interplay between sediment supply and hydrodynamic forces, including river discharge, tidal action, sea level changes, and wave energy, determines the morphology and progradation rates of deltas (Galloway, 1975; Nienhuis et al., 2020).

River deltas hold critical importance for human activities due to their fertile soils, rich biodiversity, and strategic locations. Consequently, they have been often subject to anthropogenic interventions which have significantly impacted their morphological evolution, as shown in Figure 1. Indeed, large-scale engineering projects, such as dam construction, river channelization, and sand mining, alter sediment transport dynamics, often reducing the sediment load reaching deltas (Nittrouer and Viparelli, 2014).





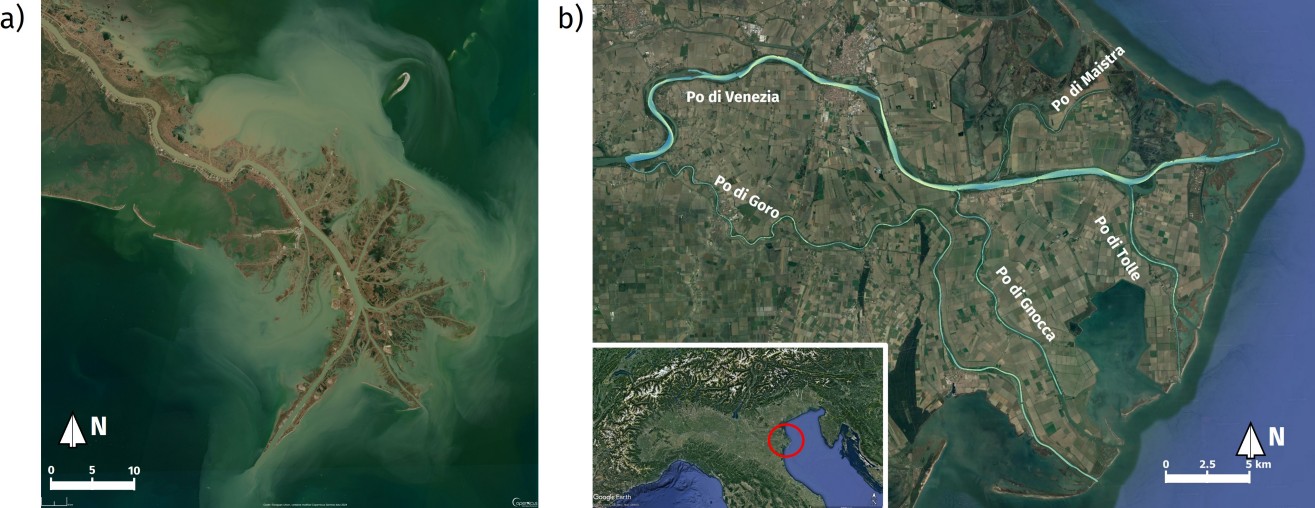

**Figure 1.** Satellite imagery depicting engineered river-dominated deltas. a) The Mississippi River Delta, Louisiana, USA (29°10′ N, 89°15′ W), showcasing the delta's "birdfoot" shape (© Copernicus Sentinel Hub). b) The Po River Delta, located in northeastern Italy (44°57′ N, 12°26′ E), where the primary distributary channel 'Po di Venezia' flows into the Adriatic Sea. A sequence of secondary distributary branches diverges from the main channel, comprising the 'Po di Goro,' 'Po di Gnocca,' 'Po di Maistra,' and 'Po di Tolle' branches, progressing downstream (© Google Earth 2024).

The Po River Delta and its basin, located in northern Italy (Figure 1b), provide an example of such a delicate system, being one of the most significant and anthropized deltas in the Mediterranean region. It drains a significant amount of water and carries a substantial quantity of sediment coming from the whole Italian Alps and the northern part of the Apennines. Similar to most rivers globally, the Po River is affected by extensive dam construction and water management practices throughout its basin, which have led to a reduction in the sediment load reaching the delta, thereby impacting its growth and stability (Syvitski

et al., 2005). Moreover, extensive land reclamation projects have progressively transformed wetlands into agricultural and urban areas, altering the natural hydrology and sediment dynamics of the delta. Specifically, one of the measures taken to this end has been the construction of levees along all distributary channels in the delta, resulting in a decreased overflow onto the plains and consequently enhancing local natural subsidence (Syvitski et al., 2009).

   Understanding the main mechanisms determining the long-term evolution of river-dominated deltas is thus crucial for their

proper management (Edmonds et al., 2022). However, most studies related to river deltas focus merely on the hydrodynamics of the deltas and nearshore environments to understand the flow routes in channels and wetlands for bio-ecological purposes (Canestrelli et al., 2010; Maicu et al., 2018). Planform numerical simulations aimed at studying the long-term morphodynamic evolution are rare and generally based on schematic delta configurations due to the high computational cost and several simplifying assumptions required in models (Overeem et al., 2005; Fagherazzi and Overeem, 2007; Edmonds et al., 2010; Guo et al.,

2015; Moodie et al., 2019).



Another branch of literature models the morphodynamic evolution of deltas in a one-dimensional framework, coupled with a quasi-two-dimensional model for the interactions among delta channels at each bifurcation (Bolla Pittaluga et al., 2003, 2015). These models allow for easier isolation of the main mechanism affecting the flow partitioning at each bifurcation (Durante et al., 2024) and chances of avulsion towards secondary branches (Salter et al., 2018; Barile et al., 2023). Despite their simplifications, and the assumption that riverbed changes are much slower than the hydrodynamic variations, they provide powerful insight into the temporal evolution of the delta channels. Interestingly, Salter et al. (2020) with a simple delta network model found that two-way coupling between upstream and downstream bifurcations in a network can lead to chaotic dynamics in the network.

Using a similar tree-like structural model, Ragno et al. (2022) examined the long-term equilibrium configuration of tide-influenced deltas. This analytical model enables rapid identification of the equilibrium solution by solving a system of equations. This approach allows for isolating and assessing differential variations within the system and their impact on flow distribution at each delta bifurcation.

Building on the work of Ragno et al. (2022), here we formulate a new model of river-dominated deltas where the system is solved as a sequence of bifurcations, each one implemented according to the recent work of Durante et al. (2024). In the latter work the Authors redefined the two-cell model by Bolla Pittaluga et al. (2003) imposing an energy balance at the bifurcation node and relaxing the assumption of free-surface elevation equality at the node. As consequence, the system of equations governing sediment and discharge partitioning at the bifurcation gains a new degree of freedom, allowing for the detection of intrinsic asymmetries related to downstream effects. This approach demonstrates that the length of the downstream branches plays a crucial role on the stability of the bifurcation. Additionally, the model is formulated in a general form to include possible asymmetries within each bifurcation, allowing for proper application to real-world scenarios.

In the present study, the capability of the model to reproduce real scenarios is tested by applying it to the Po River Delta. Whereas the morphodynamic equilibrium of the Po River upstream of the first bifurcation has been explored within the framework of a 1D model (Lanzoni et al., 2015), no attempts have been done so far to interpret the bed morphology of the delta with a morphodynamic model. The reliability of the obtained equilibrium configurations is assessed comparing model result with the discharge partitioning among the deltaic branches, provided by flow measurements collected during a series of field campaigns by the Po River Basin Authority from 2002 to 2011 (Zasso and Settin, 2012). Furthermore, model-based estimations of free-surface and bed elevation profiles are validated against cross-sectional measurements taken in 2018. This approach facilitates an assessment of the current equilibrium state of the Po River Delta and allows for the evaluation of potential impacts arising from avulsion events, offering critical insights for effective delta management.

The following sections of the paper are structured as detailed below: Section 2 provides a formulation of the analytical procedure employed in this study. Section 3 delineates the main processes involved in a general idealized delta configuration. In Section 4 we apply the model to the case of the Po River delta, comparing model results with available measurements. Finally Section 5 is dedicated to discussing the results, while Section 6 concludes the paper providing a summary of the principal observations and insights.



## 2 Formulation of the analytical model

From a modelling perspective, a delta can be formally represented as a network of interconnected bifurcations, originating from a single upstream channel referred to as the "delta apex", as depicted in Figure 2b. This structure branches progressively from the primary (first) node to the terminal nodes, where the downstream channels discharge into the sea at a defined free-surface elevation.

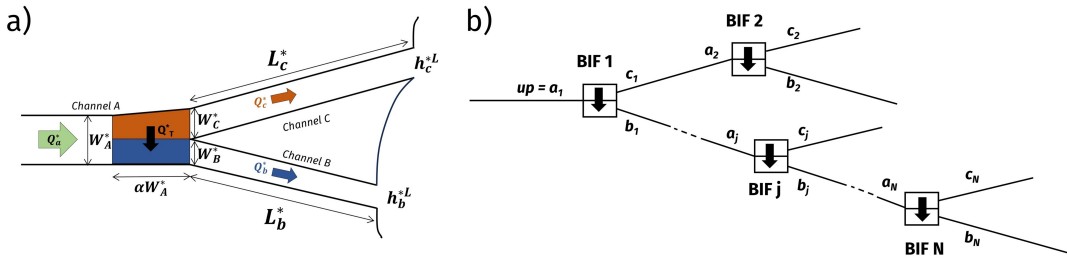

**Figure 2.** (a) Schematic representation of a single bifurcation model as proposed by Durante et al. (2024), illustrating the inclusion of multiple asymmetries. (b) Plan view of an idealized river delta, depicted as a network comprising $j$ bifurcations, where $j = 1, \ldots, N$.

Before addressing the problem of the entire delta system, it is essential to first introduce how each of its functional units, specifically a single bifurcation, is approached. As illustrated in Figure 2a, a single bifurcation consists of an upstream channel $a$, a node, and two downstream branches $b$ and $c$ respectively. The node is modelled using the two-cell model initially proposed by Bolla Pittaluga et al. (2003) and recently refined by Durante et al. (2024) to account for potential asymmetries. To this end, Durante et al. (2024) introduced the following key parameters: the length ratio between the two downstream branches, denoted as $\gamma_L$; the width ratios of each single downstream branch referred to the upstream channel, $r_b$ and $r_c$; the total downstream enlargement represented by $r_a$; the kinetic head losses at the bifurcation node $\xi$; and the downstream water level differential $\Delta h^L$. More in detail, they read:

- Length ratio: $\gamma_L = \frac{L_c^*}{L_b^*}$;

- Branch width ratios: $r_b = \frac{W_b^*}{W_a^*}$, $r_c = \frac{W_c^*}{W_a^*}$

- Downstream enlargement: $r_a = \frac{W_b^* + W_c^*}{W_a^*} = r_b + r_c$

- Downstream level asymmetry: $\Delta h^L = h_b^L - h_c^L$,

where $W_i^*$ and $L_i^*$ are the width and length of the $i^{th}$ branch respectively, with $i = \{a, b, c\}$ denoting the channel within the bifurcation, and $h_i^L$, with $i = \{b, c\}$, is the free-surface elevation at the downstream end of the $i^{th}$ branch made dimensionless with the uniform flow depth of the upstream channel.

The model does not consider temporal variations in the geometry, implying no variation in each width and length over time. A uniform grain size $d_s^*$ is assumed to compose the river bed. The bifurcation is fed upstream with a constant discharge $Q_a^*$ and





a constant sediment flux $Qs_a^*$ in equilibrium with local hydrodynamics. Furthermore, the model assumes rectangular straight channels, allowing for a unique value for the bed slope $s_i$ for the $i^{th}$ channel. Consequently, the free-surface elevations at the node $h_i^{*N}$ can be calculated from the downstream elevation $h_i^{*L}$ given the branch length $L_i^*$.

Note that, for a single bifurcation the free-surface elevation $h_i^{*L}$ at the downstream end of the $i^{th}$ channel branch is imposed as an input parameter and typically set to a fixed value (such as mean sea level).

The model is grounded in a nodal point condition that considers transverse flow and sediment exchange between the two cells in the upstream channel $a$, influencing the flow partitioning in the downstream branches $b$ and $c$. The transverse solid exchange is evaluated based on the procedure for describing two-dimensional bed load transport over an inclined bed (Ikeda et al., 1981):

$$q_{Ts}^* = q_{as}^* \left[ \frac{V^*}{\sqrt{U^{*2} + V^{*2}}} - \frac{r}{\sqrt{\vartheta_a}} \frac{\partial \eta^*}{\partial y^*} \right], \tag{1}$$

where dimensional variables are denoted with the superscript $^*$. Specifically, $q_{Ts}^*$ represents the dimensional transverse solid discharge per unit width, and $q_{as}^*$ denotes the longitudinal solid flow discharge per unit width from upstream. $U^*$ and $V^*$ are the longitudinal and transverse velocity components, respectively. $\partial \eta^* / \partial y^*$ denotes the transverse bed slope at the junction of channels $b$ and $c$. The parameter $r$ is an experimental constant that ranges between $0.3$ and $1$ (Ikeda et al., 1981; Talmon et al., 1995). The Shields parameter, $\vartheta_a$, associated with the uniform flow is defined as:

$$\vartheta_a = \frac{Q_a^{*2}}{\Delta g d_s^* C_a^2 D_a^{*2} W_a^{*2}} \tag{2}$$

where $\Delta$ is the submerged sediment density ($\Delta = \rho_s/\rho - 1$, with $\rho_s$ and $\rho$ the densities of sediment and water, respectively), $g$ is the gravitational acceleration, $C_a$ is the dimensionless Chézy coefficient, $D_a^*$ is the flow depth, and $W_a^*$ is the channel width.

After having appropriately made dimensionless the variables of the problem with respect to the quantities in the upstream channel $a$, as in Durante et al. (2024), and following algebraic manipulation (details in Appendix A), the nodal point condition is reformulated as:

$$q_{Ts} = \frac{q_T}{D_{abc}U_{abc}} - \frac{r}{\beta\sqrt{\vartheta_a r_a}} \left( h_b^N - h_c^N - D_b + D_c \right), \tag{3}$$

where $D_{abc}$ and $U_{abc}$ represent the dimensionless average water depth and flow velocity at the node, respectively. Additionally, $h_i^N$ and $D_i$ are the dimensionless free-surface elevation and the water depth at the node in the $i^{th}$ channel, and $\beta$ is the upstream channel aspect ratio defined as:

$$\beta = \frac{W_a^*}{2D_a^*}. \tag{4}$$

To close the problem and find the equilibrium solutions of the bifurcation, we refer to the procedure employed by Durante et al. (2024), where the system of necessary equations to find the equilibrium solutions is determined by five equations and five unknowns (namely, the flow discharge and water depth in every branch and the free-surface elevation at the node). Interestingly,





employing an energy balance approach at the node, the system gains a new degree of freedom allowing for a slight disparity in water levels at the node, which might allow unequal flow velocities.

As demonstrated by Durante et al. (2024), the key factors influencing bifurcation stability include the dimensionless length of the branches $L$ and the bifurcation parameter $R$, in line with previous studies (Ragno et al., 2020, 2022; Salter et al., 2018):

$$L = \frac{L^* s_a}{D_a^*} \quad R = \frac{\alpha r}{\beta \sqrt{\vartheta_a}}, \tag{5}$$

where $L$ is the branch length $L^*$ scaled with the backwater length, and $\alpha$ is the dimensionless length of the upstream channel as defined by Bolla Pittaluga et al. (2003).

Let's now extend the formulation to the case of a delta. As previously introduced, a delta can be described as a series of interconnected bifurcations, each considered an individual unit governed by its own set of equations. Unlike the scenario of a single bifurcation, the equations governing each node within the delta are interdependent with those of adjacent bifurcations.

This coupling necessitates solving simultaneously the system of equations for all bifurcations to achieve equilibrium for the entire deltaic network.

Assuming that the system is fed upstream with a constant discharge $Q_{up}^*$ and a constant sediment flux $Qs_{up}^*$ in equilibrium with local hydrodynamics, in a similar fashion as Durante et al. (2024), the equations are made dimensionless by scaling the dimensional variables with reference to the dimensions of the 'delta apex' channel:

$$(D_{ij}, h_{ij}) = \frac{(D_{ij}^*, h_{ij}^*)}{D_{up}^*}, \quad (q_{ij}, q_{Tj}) = \frac{(q_{ij}^*, q_{Tj}^*)}{q_{up}^*}, \quad (q_{isj}, q_{Tsj}) = \frac{(q_{isj}^*, q_{Tsj}^*)}{q_{s_{up}}^*}, \tag{6}$$

where the subscript $up$ denotes the 'delta apex' upstream channel, while the subscript $i$, with $i = \{a, b, c\}$ denotes the position of the channel within each $j^{th}$ bifurcation: $i = a$ denotes the upstream channel, while $i = b, c$ indicate the two downstream channels, as illustrated in Figure 2.

The mathematical formulation for each $j^{th}$ bifurcation follows the approach outlined by Durante et al. (2024), which incorporates the necessary system of five equations governing the flow and sediment conservation, the energy balance, and the

nodal condition:

   1. Flow discharge balance:

$$q_{b_j} r_{b_j} + q_{c_j} r_{c_j} = q_{a_j} \tag{7}$$

   2. Solid discharge balance:

$$q_{bs_j} r_{b_j} + q_{cs_j} r_{c_j} = q_{as_j} \tag{8}$$

   3. Energy balance in cell b:

$$h_{b_j}^N + (1 + \xi_j) \frac{Fr^2}{2} \frac{q_{b_j}^2}{D_{b_j}^2} = h_{a_j}^N + \frac{Fr^2}{2} \frac{q_{a_j}^2}{D_{a_j}^2} \tag{9}$$



4. Energy balance in cell c:

$$h_{c_j}^N + (1 + \xi_j)\frac{Fr^2}{2}\frac{q_{c_j}^2}{D_{c_j}^2} = h_{a_j}^N + \frac{Fr^2}{2}\frac{q_{a_j}^2}{D_{a_j}^2} \tag{10}$$

5. Nodal condition:

$$\frac{q_{bs_j}}{q_{as_j}}r_{a_j} - 1 = \frac{q_{b_j}r_{a_j} - q_{a_j}}{D_{abc_j}U_{abc_j}} + \frac{R_j}{r_{b_j}}\left[h_{b_j}^N - h_{c_j}^N - D_{b_j} + D_{c_j}\right], \tag{11}$$

The Froude Number $Fr$ in the upstream channel is defined as:

$$Fr = \frac{q_{up}^*}{\sqrt{gD_{up}^{*3}}}. \tag{12}$$

Noteworthy, the local approach adopted here needs a careful definition in each bifurcation of the asymmetry parameters (i.e. $\gamma_{L_j}$, $r_{a_j}$, $r_{b_j}$, $r_{c_j}$ and $\xi_j$), the average water depth and flow velocity at the node ($D_{abc_j}$ and $U_{abc_j}$ respectively) and most importantly the bifurcation parameter $R_j$ (where $\beta$ and $\vartheta_a$ are locally evaluated).

The nodal free-surface elevation for channel $b$ and $c$ are determined taking advantage of the normal flow equation in a rectangular channel at the equilibrium as:

$$h_{i_j}^N = h_{i_j}^L + L_{i_j}\frac{q_{i_j}^2 C_{up}^2}{D_{i_j}^3 C_{i_j}^2}, \tag{13}$$

Note that, while for a single bifurcation, the free-surface elevation $h_i^L$ at the downstream end of the $i^{th}$ channel branch is imposed as an input parameter, when extending the analysis to the case of a delta, it still remains known only for the terminal branches (ending order bifurcations). Importantly, in internal bifurcations, the downstream free-surface elevation $h_{i_j}^L$

is a variable of the system, since it is set equal to the nodal level of channel $a$ in the downstream bifurcation $h_{a_{j+1}}^N$. Therefore, the downstream water level asymmetry $\Delta h^L$ in each bifurcation is not predetermined and is instead linked to the local length ratio $\gamma_L$.

Given that the delta system consists of multiple interconnected bifurcations, solving the model requires addressing $5N$ equations, where $N$ is the number of bifurcations. The complexity of the system increases as the number of bifurcations grows,

as each bifurcation influences downstream conditions and, conversely, is affected by upstream bifurcation outcomes. Flow partitioning at an upstream bifurcation dictates the flow distribution to subsequent branches, while free-surface elevations at downstream bifurcations feed back into the system, modifying the hydraulic conditions at earlier bifurcations.

The system allows for three potential equilibrium states at each bifurcation: (i) dominance of one branch, (ii) dominance of the other branch, or (iii) symmetric partitioning between branches. Consequently, for a delta system with $N$ bifurcations, there

are $3^N$ possible equilibrium configurations, each representing a unique distribution of flow and sediment throughout the delta.

This exponential growth in the number of possible equilibrium solutions highlights the intricate nature of deltaic bifurcation systems. A comprehensive analytical framework is therefore required to capture all potential equilibrium states and to understand their impact on the delta's long-term morphological and hydrodynamic evolution.





## 3 Competing Effects on the Delta Equilibrium

Branch length has been identified as a critical factor influencing the stability of river bifurcations. Shorter branches tend to result in more balanced configurations, whereas longer branches can lead to highly unbalanced bifurcations or even increase the likelihood of avulsion (Salter et al., 2018; Barile et al., 2023; Durante et al., 2024).

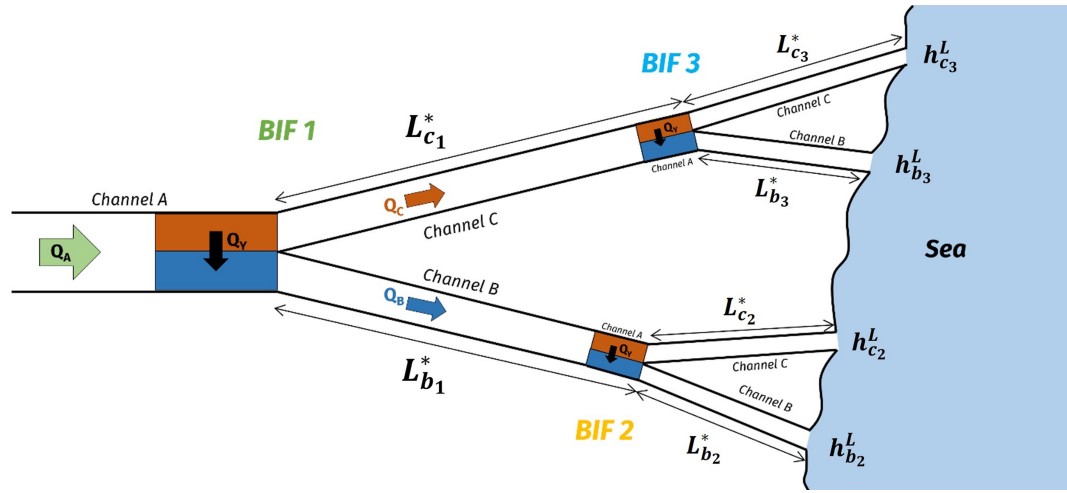

**Figure 3.** Schematic representation of an idealized river delta debouching into the sea in plan view. The delta structure comprises a primary apex bifurcation (*BIF1*) followed by secondary bifurcations downstream of each branch (*BIF2* and *BIF3*).

When considering the entire delta system, the cumulative length of branches downstream of each bifurcation is a key factor in determining the system's equilibrium state. Additionally, the equilibrium of each bifurcation is affected by the distribution of these branch lengths among the downstream bifurcations. This implies that even if the total downstream length remains constant, the specific distribution of lengths among branches can result in slight variations in flow partitioning. In a symmetrical delta with two second-order bifurcations, as sketched in Figure 3, we denote $L_{up}^*$ as the dimensional lengths of the primary branches of the apex bifurcation ($L_{up}^*=L_{b1}^*=L_{c1}^*$), and $L_{down}^*$ as the lengths of branches in downstream bifurcations ($L_{down}^*=L_{b2}^*=L_{c2}^*=L_{b3}^*=L_{c3}^*$). The aggregate length is defined as:

$$L_{tot}' = L_{up}' + L_{down}' = \left( L_{up}^* + L_{down}^* \right)/W_{up}^*, \tag{14}$$

where the length is made non-dimensional by the upstream channel width of the apex bifurcation $W_{up}^*$.

Figure 4 elucidates these dynamics, showing equilibrium solutions for symmetrical deltas as the aggregate branch length varies. Results are reported varying the upstream bifurcation parameter $R_{up}$ in terms of discharge asymmetry $\Delta Q$ between the branches in the apex bifurcation as follows:

$$\Delta Q = \frac{Q_{b1} - Q_{c1}}{Q_{up}}, \tag{15}$$

where $Q_{b1}$ and $Q_{c1}$ are the discharges in the two primary branches, and $Q_{up}$ is the discharge upstream of the apex bifurcation.

 

Panel 4a shows the solutions when varying $L^*_{up}$ while keeping $L^*_{down}$ constant, whereas panel 4b shows the solutions when $L^*_{up}$ is constant and $L^*_{down}$ is varied. Panels 4c and 4d provide a schematic representation of the delta configurations for elongated branch lengths.

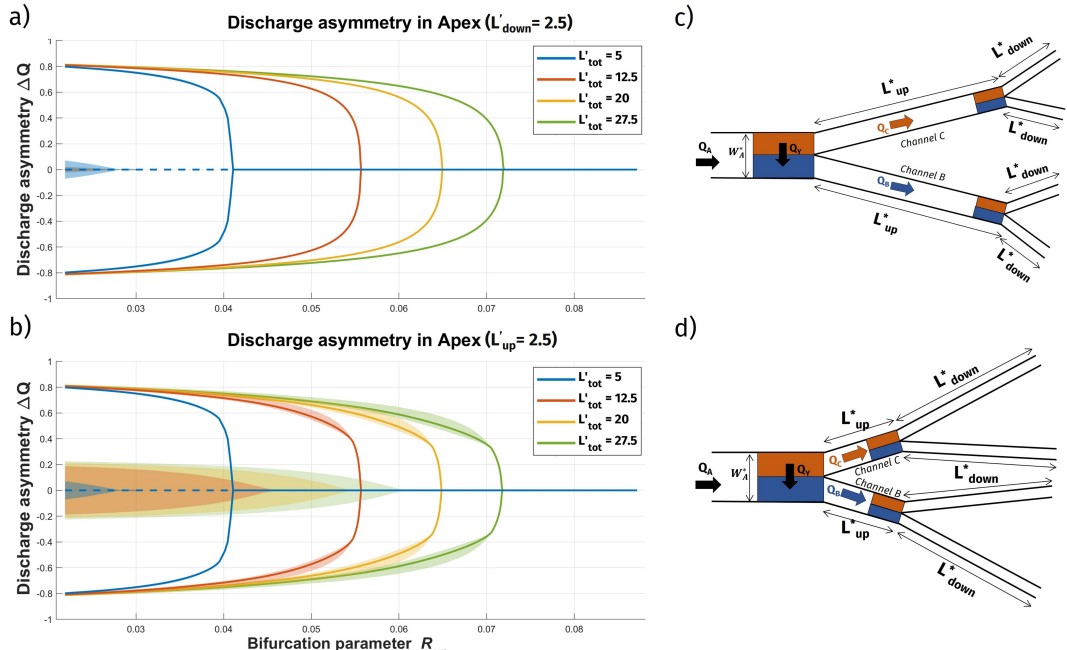

**Figure 4.** Equilibrium solutions for symmetrical deltas as the aggregate branch length varies. Panels (a) and (b) display the discharge asymmetry $\Delta Q$ at the apex bifurcation, plotted against the bifurcation parameter $R_{up}$ for different total branch lengths $L'_{tot}$. Panel (a) presents solutions varying the primary branch lengths $L^*_{up}$ (with $L'_{down} = 2.5$), while panel (b) shows solutions varying the downstream branch lengths $L^*_{down}$ (with $L'_{up} = 2.5$). Each line represents the average of all the corresponding solutions for different $L'_{tot}$ values. The shaded area indicates the range of possible solutions for the corresponding colour. Panels (c) and (d) depict the schematic configuration of the delta in the two cases. (Parameters: $\gamma_L = 1$, $r_a = 1$, $r_b = 0.5$, $\xi = 0$)

In general, for a given value of $L'_{tot}$, if $R_{up}$ is higher than a critical threshold, flow is equally distributed at the bifurcation (the system admits just one possible symmetrical solution), on the contrary, when $R_{up}$ is lower than the critical threshold

the balanced solution becomes unstable and the system attains a stable state where one of the two branches captures most of discharge (the system admits two equivalent solutions depending on which of the two downstream branches dominates).

It is clear that total branch length $L'_{tot}$ is a crucial factor in determining the behaviour of the main apex bifurcation. Specifically, shorter total branch length increase the number of configurations (values of $R_{up}$) that achieve equal partitioning of flow between the branches ($\Delta Q = 0$). This suggests that, as the total branch length decreases, the system tends toward a more

balanced state, reducing the likelihood of unbalanced flow distributions.





An intriguing observation is that, for any given value of $L'_{tot}$, the distribution of branch lengths between the primary and second-order bifurcation does not influence the critical bifurcation parameter $R_{cr}$, where unbalanced solutions first appear. This suggests that the onset of flow asymmetry is controlled mainly by the overall length of the channel network (measured from the uppermost node to the outlet) rather than by the specific distances of the secondary bifurcation nodes from the outlet.

In other words, it is the total upstream-to-outlet distance that plays a critical role in triggering flow imbalances, rather than how it is distributed among the downstream channels through the delta network.

A further significant finding can be drawn from panel (b) in Figure 4, highlighting the significance of the shaded areas. When the primary branches ($L^*_{up}$) are relatively short, increasing the length of the second-order downstream branches ($L^*_{down}$) results in slight deviations among the solutions. Specifically, when the downstream branches remain short, the complete set of $3^N$ potential solutions effectively collapses to align with the three fundamental solutions, as if the apex bifurcation were solved in isolation (depicted in panel (a) of Figure 4). However, as $L^*_{down}$ increases, an induced downstream-to-upstream feedback alters the upstream hydraulic conditions, permitting a range of discharge asymmetry values ($\Delta Q$).

This phenomenon occurs because modifications in the lengths of $L^*_{down}$ influence the hydraulic conditions at the downstream bifurcation nodes (i.e., nodes 2 and 3), which subsequently impact the water surface elevations. These alterations propagate upstream, affecting the free-surface slopes in the primary bifurcation branches and resulting in a spectrum of possible discharge asymmetry outcomes at the apex.





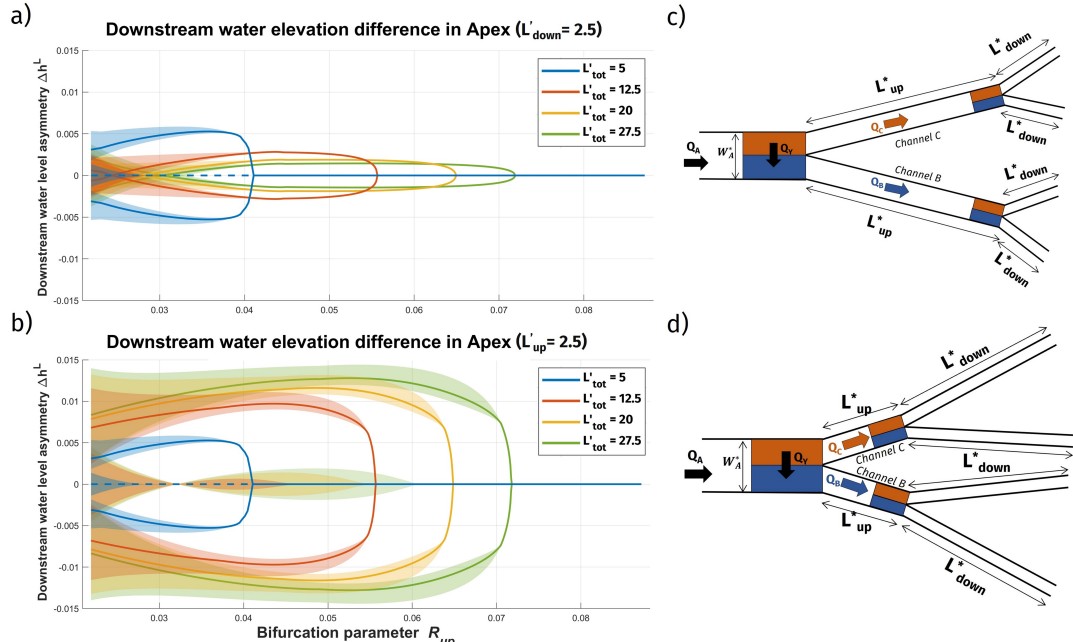

**Figure 5.** Variations of the parameter $\Delta h^L$ for symmetrical deltas as the aggregate branch length varies. Panels (a) and (b) show the downstream water level asymmetry parameter $\Delta h^L$ at the apex bifurcation plotted against the bifurcation parameter $R_{up}$ for different total branch lengths $L'_{tot}$. Panel (a) presents solutions varying the primary branch lengths $L^*_{up}$ (with $L'_{down} = 2.5$), while panel (b) shows solutions varying the downstream branch lengths $L^*_{down}$ (with $L'_{up} = 2.5$). Each line represents the values of $\Delta h^L$ for different $L'_{tot}$ values, with the shaded area indicating the range of possible values for the corresponding colour. Panels (c) and (d) depict the schematic configuration of the delta in the two cases. (Parameters: $\gamma_L = 1$, $r_a = 1$, $r_b = 0.5$, $\xi = 0$)

Figure 5 illustrates the downstream water level asymmetry parameter $\Delta h^L_1$ as a function of the bifurcation parameter $R_{up}$ for different values of the total delta length $L'_{tot}$, corresponding to the equilibrium solution presented in Figure 4. This parameter, which measures the differences in water elevation between the second-order nodes, is essential for understanding the

hydrodynamic interactions within the delta.

In particular, panel (a) in Figure 5 illustrates the case when the variation in $L'_{tot}$ is achieved varying the primary branch lengths $L^*_{up}$ and keeping $L^*_{down}$ constant, while panel (b) refers to the opposite case, i.e., varying the downstream branch lengths $L^*_{down}$ and keeping $L^*_{up}$ constant.

The comparison between Figures 5 (a) and (b) shows that lengthening of downstream branches leads to an increase in $\Delta h^L$,

thereby exerting a greater influence on the discharge partitioning at the upstream bifurcation.

However, river deltas rarely exhibit purely symmetric structures. Complex internal feedback mechanisms during delta formation and evolution typically produce variations in branch lengths and widths (Fagherazzi et al., 2015). These asymmetries are shaped by multiple factors, including planform geometric constraints, vegetation growth, and variations in downstream





conditions. Such influences create differential sediment supply across branches, leading to uneven land construction and, con-

sequently, differential channel lengthening and widening.

In Figure 6, the impact of differential branch length at the apex bifurcation is analyzed by varying the total length ratio $\gamma_L$ which is thus defined as $L'_{c_{tot}}/L'_{b_{tot}}$. In order to isolate this effect the second-order bifurcations are kept symmetrical and no width enlargement or differential width of primary branches is included.

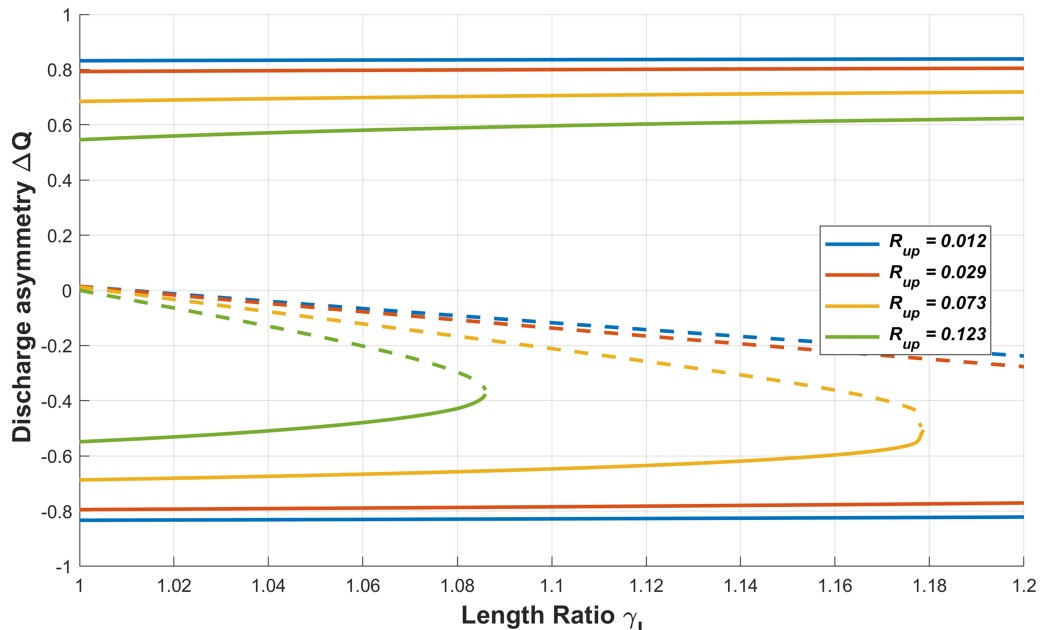

**Figure 6.** Discharge asymmetry in the apex bifurcation as a function of the total differential branch length ratio $\gamma_L$. The plot shows the variation of $\Delta Q$ with respect to $\gamma_L$ for four different values of the upstream bifurcation parameter $R_{up}$, depicted with the corresponding colour (other parameters are $r_b = 0.5$, $r_a = 1$, $L'_{tot} = 90$, $\xi = 0$).

An increase in $\gamma_L$ results in branch $c$ attaining a greater length relative to branch $b$, thereby enhancing the flow partition in

favor of branch $b$. Conversely, intermediate solutions and those favoring branch $c$ become less favorable with increasing $\gamma_L$. This occurs because longer branches are associated with a lower free-surface gradient, which reduces their sediment transport capacity and promotes sediment accumulation. Moreover, for a given value of $R_{up}$, if the length ratio is sufficiently large, the dominance of the shorter branch becomes the only possible solution. Therefore, there exist threshold conditions for high values of $\gamma_L$ and $R_{up}$ where no switch in dominance can be found.

Another characteristic feature of bifurcations in river deltas is the differential width partition between branches. Typically, the dominant branch in natural deltas exhibits a larger width compared to the branch with lower flow. This geometrical feature is incorporated in the model through the branch width ratio $r_b$.





For symmetrical lengths and no width enlargement at the apex bifurcation (i.e., $\gamma_L = 1$ and $r_a = 1$), defining $r_b$ as $W_{b1}/W_{up}$, increasing $r_b$ signifies widening the primary branch $b$ and narrowing the other branch. In Figure 7, this effect is isolated while maintaining symmetrical bifurcations downstream.

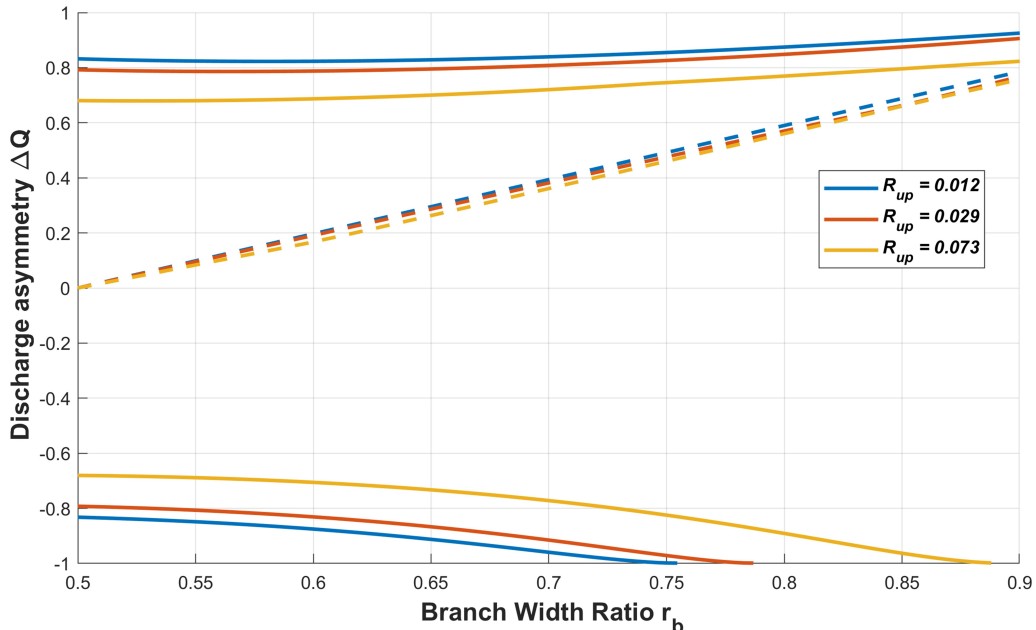

**Figure 7.** Discharge asymmetry in the apex bifurcation as a function of the branch width ratio $r_b$. The plot shows the variation of $\Delta Q$ with respect to $r_b$ for three different values of the upstream bifurcation parameter $R_{up}$, depicted with the corresponding colour (with parameters $\gamma_L = 1, r_a = 1, L'_{tot} = 50, \xi = 0$).

Figure 7 illustrates the variations in $r_b$ from symmetry (i.e., $r_b = r_a/2 = 0.5$) to a highly asymmetrical state where branch $b$ attains 90% of the upstream width, across three different values of the upstream bifurcation parameter $R_{up}$. For the dominant branch, the discharge partitioning remains relatively unaffected by variations in $r_b$. However, the intermediate unstable solution shows an increase in discharge through branch $b$ as $r_b$ increases, eventually achieving almost the same partitioning as the dominant branch solution. More interestingly, for the scenario where the narrower branch is dominant, the system tends towards complete closure of the larger branch as $r_b$ increases. Beyond a certain threshold value of $r_b$, the solution that advantages the narrower branch ceases to exist, indicating that for high values of $r_b$, only solutions where the larger branch is dominant are feasible. The bifurcation parameter $R_{up}$ does not significantly impact the results, except that higher values of $R_{up}$ are associated with less unbalanced configurations.



## 4   Equilibrium Configurations of the Po River Delta

The Po River is the Italy's longest river. It extends over $652\,\mathrm{km}$ and drains an area of approximately $71,000\,\mathrm{km}^2$. Flowing from west to east across northern Italy, the river ultimately discharges into the Adriatic Sea through a delta located several kilometers south of the Venice Lagoon. Within its deltaic system, the Po River's main channel, known as the Po di Venezia, undergoes four morphodynamically active bifurcations, generating four distributary branches as illustrated in Figure 1b. Specifically, the main channel successively encounters the Po di Goro, Po di Gnocca, Po di Maistra, and Po di Tolle bifurcations as it progresses downstream. Prior to reaching the Adriatic Sea, the primary river channel terminates in three distinct mouths within the Po Delta Lagoons, namely Busa di Dritta, Busa di Tramontana, and Busa di Scirocco.

The Po di Venezia carries the majority of the upstream discharge under all flow conditions and, to accommodate this, it attains a larger width compared to the secondary branches. The average width and length of each channel in the delta are detailed in Table 1, while the average width of the upstream channel at the apex is $370\,\mathrm{m}$.

**Table 1.** Dimensions of channels in the Po Delta. The upper section lists the secondary branches, while the lower section presents the channels composing the main stem (Po di Venezia).

| Channel | Width [m] | Length [km] |
|---|---|---|
| Po di Goro | 95 | 48.4 |
| Po di Gnocca | 125 | 21.1 |
| Po di Maistra | 60 | 17.1 |
| Po di Tolle | 180 | 10.9 |
| Goro - Gnocca | 380 | 28.9 |
| Gnocca - Maistra | 400 | 1.9 |
| Maistra - Tolle | 420 | 8.6 |
| Tolle - Sea | 400 | 9.3 |

The local asymmetry parameters for each bifurcation have been quantitatively assessed, with the respective values provided in Table 2. Notably, a downstream enlargement is evident at each bifurcation within the delta, consistent with the data reported by Barile et al. (2023b). Furthermore, the branch width ratio $r_b$ exhibits significant asymmetry across all bifurcations, with the most pronounced asymmetry occurring at the Maistra bifurcation.



**Table 2.** Asymmetry parameters for the bifurcations in the Po Delta.

| Channel | $r_a$ | $r_b$ | $\gamma_L$ |
|---|---|---|---|
| Po di Goro | 1.28 | 1.03 | 1.67 |
| Po di Gnocca | 1.38 | 1.05 | 11.1 |
| Po di Maistra | 1.20 | 1.05 | 1.99 |
| Po di Tolle | 1.38 | 0.95 | 1.17 |

The delta model employed in this study is calibrated to the current geometric configuration of the Po River Delta down to the Tolle bifurcation. Beyond this point, the model does not reliably capture the morphodynamics of the Po Delta Lagoons. As such, downstream of the Tolle bifurcation, the main stem is modeled as debouching primarily through the central mouth, i.e., the Busa di Dritta. Therefore, given the sequence of four bifurcations, the system theoretically offers 81 potential equilibrium solutions.

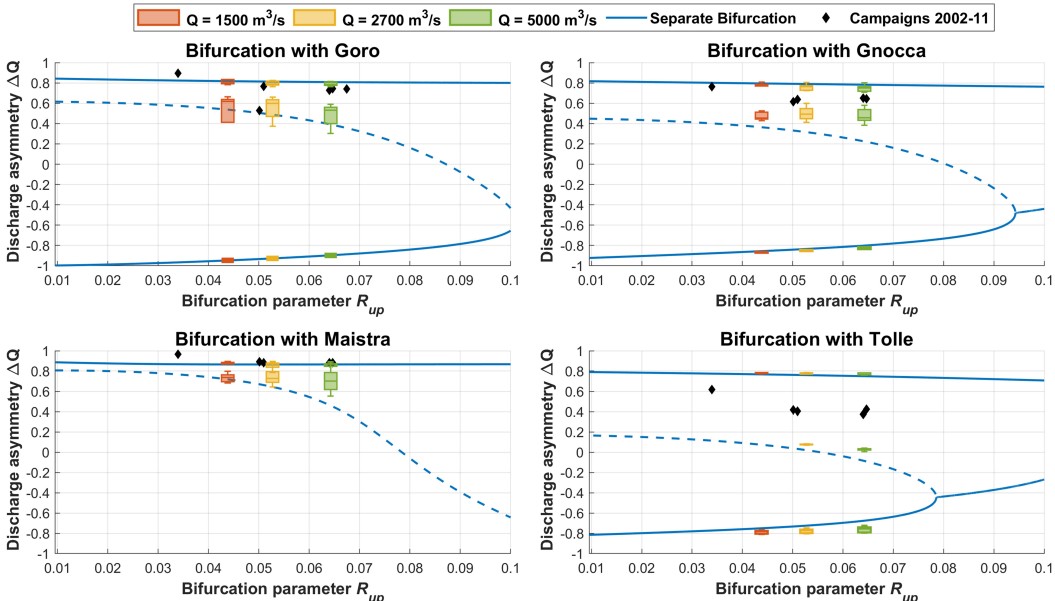

**Figure 8.** Equilibrium discharge asymmetry $\Delta Q$ as a function of bifurcation parameter $R_{up}$ for each bifurcation in the Po Delta (Goro, Gnocca, Maistra, and Tolle), listed in order from the apex progressing downstream. Boxplots represent model results at upstream discharges of 1500 m$^3$/s (orange), 2700 m$^3$/s (yellow), and 5000 m$^3$/s (green). Solid and dashed blue lines indicate predictions from the separate bifurcation model by Durante et al. (2024). Field data from 2002–2011 are marked by black diamonds.

The equilibrium solutions are reported in Figure 8 in terms of discharge asymmetry $\Delta Q$ displayed for each bifurcation progressing downstream from the apex bifurcation. Model outputs are depicted using boxplots to illustrate the variability of





each equilibrium solution due to internal feedback mechanisms within the deltaic system. Results are provided for three distinct upstream discharge values $Q^*_{up}$ (or $R_{up}$), differentiated by color within the boxplots to represent the flow distribution under conditions representative of formative, reduced, and elevated discharges of the delta (2700 $\mathrm{m}^3/\mathrm{s}$, 1500 $\mathrm{m}^3/\mathrm{s}$ and 5000 $\mathrm{m}^3/\mathrm{s}$,
respectively).

Additionally, leveraging the prior finding that the overall branch length serves as a primary determinant of equilibrium configurations across delta bifurcations, results from this study are compared with those of the single-bifurcation model proposed by Durante et al. (2024). To facilitate this comparison, internal feedback mechanisms are disregarded, and the branches are assumed to discharge directly into the sea. Therefore, the length of each bifurcation's branches is calculated by adding the
average length of the downstream branches to the corresponding length of the Po di Venezia reported in the lower section of Table 1. Despite the necessary simplifications, both models demonstrate good alignment with observed discharge partitioning data for the Po Delta from 2002 to 2011 (Zasso and Settin, 2012).

Interestingly, neither the delta model nor the separate bifurcation model can produce equilibrium configurations that exhibit dominance of the Maistra branch (i.e. $\Delta Q < 0$). This result is consistent with the high asymmetry of this bifurcation, as
indicated in Table 2 and illustrated by threshold conditions in Figure 7.

Notably, at the downstream-most bifurcation with Po di Tolle, model results show less agreement with field measurements. This discrepancy is likely due to the model's simplification in representing only one of the three main distributary mouths of the Po di Venezia, omitting the additional discharge redistribution across these channels and the associated delta lagoons. These complex hydro-morphodynamic interactions are not included in the current model formulation.





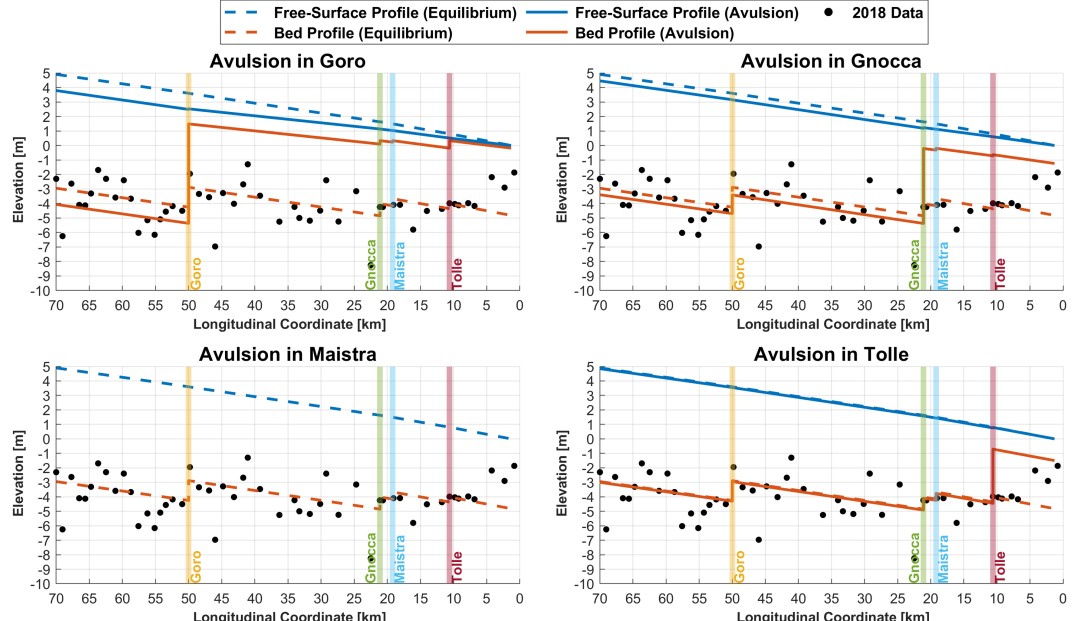

**Figure 9.** Equilibrium bed and free-surface profiles along the Po Delta main stem (Po di Venezia) under the formative discharge of 2700 m$^3$/s, with avulsion scenarios at each bifurcation (Goro, Gnocca, Maistra, and Tolle). Dashed lines indicate equilibrium profiles; solid lines show profiles post-avulsion. Black dots represent cross-sectional bed elevations from 2018. Colored vertical lines mark bifurcation location.

The model enables delineation of free-surface and bed profiles along all delta channels. Considering the formative discharge upstream of 2700 m$^3$/s, it predicts the equilibrium profiles that conform to the current delta geometry. Figure 9 shows the equilibrium profile of the Po Delta main stem (Po di Venezia), alongside cross-sectional averaged bed level data from 2018. Despite variability due to natural channel width fluctuations, the model accurately captures bed slopes in each channel, evaluating the flow discharge diverted into the secondary branches (whose location is represented with colored vertical lines). Notably, the
seaward 5 km are not accurately captured, since the flow is dispersed across all the mouths and the hypothesis of constant width channel is certainly not met.

Since the equilibrium analysis includes all configurations with potential dominance shifts at each bifurcation, each subplot of Figure 9 also shows bed profiles for hypothetical avulsion events at each bifurcation. As previously discussed, the avulsion in the Po di Maistra was never found within this framework.

The switch in dominance towards secondary branches would lead to a substantial deposition in the main stem, thereby affecting navigability downstream. In contrast, significant erosion is anticipated in the secondary branch to accommodate the increased discharge. Furthermore, avulsions in any branch would alter the free-surface slope, consequently impacting the elevation at the upstream boundary. Notably, an avulsion in the Po di Goro branch could lower the free-surface elevation at the bifurcation node by approximately one meter, with potential upstream consequences resulting from the associated bed erosion.





## 5 Discussions

River deltas are subject to a complex array of dynamic forcings that exhibit continuous variability. Both upstream discharge and downstream water levels are increasingly influenced by extreme low and high conditions, a trend that is being exacerbated by climate change. In addition, the morphology of river deltas is constantly evolving due to a combination of anthropogenic interventions and natural processes. These include channel widening resulting from bank collapse and modifications in channel length driven by variations in sediment supply. Vegetation also plays a critical role in stabilizing banks and mouth bars within prograding deltas, primarily through root systems and the trapping of suspended sediments.

Given these dynamic conditions, the concept of long-term morphodynamic equilibrium in river deltas may be inherently transient. Nonetheless, the model presented in this study offers simplicity and computational efficiency, facilitating the exploration of key factors that influence flow and sediment distribution at each bifurcation. This model also enables the prediction of evolutionary trends for a given deltaic configuration. Variations in input parameters or planform geometry can be easily incorporated into the model, allowing for an assessment of their impacts on the system's equilibrium. Consequently, the evolutionary trajectory of the delta can be interpreted as a sequence of quasi-equilibrium states.

The analysis of idealized river deltas provides insights into the fundamental mechanisms underlying bifurcation evolution. By isolating specific geometric variations or asymmetries inherent in natural deltas, the model helps identify the key factors controlling discharge partitioning and the potential formation of bed-level differences at each bifurcation.

As widely recognized in the literature (Salter et al., 2018; Ragno et al., 2020; Barile et al., 2023; Durante et al., 2024), the length of the branches is a critical determinant in establishing the equilibrium solution of riverine bifurcations. However, in the context of river deltas, the sequential nature of bifurcations introduces additional complexities. These complexities complicate the estimation of the equilibrium configuration, which cannot be determined solely based on the overall branch lengths but must also account for internal asymmetries within the delta.

Our findings indicate that the total branch length, $L'_{tot}$, plays a critical role in determining the extent of flow partitioning imbalance between branches. Shorter branches tend to result in a more balanced flow distribution. Intriguingly, regardless of the length distribution in downstream bifurcations, the critical conditions distinguishing between balanced and unbalanced upstream bifurcations are primarily governed by $L'_{tot}$. The fact that the distribution of downstream branch lengths does not affect the critical bifurcation parameter $R_{cr}$ suggests that computing single bifurcations separately, as if they directly discharge into the sea, may offer a faster yet reliable prediction method.

However, feedback between bifurcations remains significant. As illustrated in panel 4b, the possible equilibrium configurations are influenced by differences in downstream water levels at the primary branches. Figure 5 depicts the downstream water level asymmetry parameter, $\Delta h^L$, at the apex bifurcation for each equilibrium solution shown in Figure 4. Notably, when the primary branches are relatively short, an increase in $\Delta h^L$ directly impacts the upstream bifurcation. In other words, the hydraulic gradient in the primary branches is affected by downstream conditions, leading to slight variations in flow distribution. As also found by Edmonds and Slingerland (2008), the water elevation at the bifurcation can originate a backwater effect that could propagate the perturbation further upstream in the delta.





It is important to note that these considerations are influenced by the assumption of uniform flow in the branches, particularly
when width variations are not accounted for. The free-surface profile is significantly affected by flow conditions, leading to
potential variations in water elevation at the nodes, especially when branches are short. This factor will be addressed in future
research.

Despite the intrinsic internal feedback being intriguing, naturally forced planimetric asymmetries exert a higher influence on
the overall evolution of the system. The specific evolutionary dynamics during each bifurcation formation within river deltas
are highly influenced by external forcings or planimetric constraints, which lead to asymmetrical branch lengths and widths.

The results presented in Figure 7 yield significant insights: when the width differential between the downstream branches
is constrained, solutions exist where both branches may exhibit dominance. However, as the differential increases (i.e., as
$r_b$ approaches $r_a$), the system converges to a single solution in which the wider branch conveys the majority of the flow.
This phenomenon could serve as a foundation for explaining channel abandonment in naturally evolving deltas. In particular,
conditions may develop where one branch widens at the expense of the other, which narrows due to sediment deposition.
Under steady-state conditions, the model indicates that beyond a critical threshold for $r_b$, the narrowing branch loses its ability
to dominate the flow. Nevertheless, the model does not capture the influence of potential external triggering events that could
induce abrupt geometric changes in the system, leading to modified hydrodynamic regimes and alternative flow configurations.

A similar behavior is observed when there is a differential in branch length. Generally, the shorter branch carries more water
relative to its longer counterpart due to a higher gradient in the free surface. However, when the length differential is not
substantial (i.e., for values of $\gamma_L$ close to 1), instances where the longer branch dominates may still occur, as depicted in Figure
6. Nonetheless, above a certain threshold $\gamma_L$, based on the characteristic parameters of the system, the only possible solution
is one where the shorter branch dominates.

This mechanism underlies river delta avulsion: during the early stages of delta formation and subsequent progradation, as
new bifurcations emerge, flow is distributed relatively evenly across the branches, maintaining a radially symmetric deltaic
structure. As long as the length differential is limited, both branches of a bifurcation may alternately carry a higher discharge.
Even in prograding branches, Salter et al. (2018) observed that internal feedbacks can sustain alternating dominance between
branches, a process known as 'soft avulsion'. However, as branches elongate, they become less responsive to hydrodynamic
variations, amplifying the length differential and leading to the dominance of the shorter branch, potentially culminating in the
complete closure of the longer one.

The intriguing observations obtained with the presented delta model, lead us to question whether multiple solutions may exist
in real-world deltas. The application to the Po River Delta, Italy, allowed us to estimate the current equilibrium configuration.

The system, characterized by a sequence of four bifurcations, theoretically permits up to 81 possible equilibrium states.
However, due to the pronounced geometric asymmetries detailed in Table 2, many of these theoretical states are unattainable
within the Po River Delta. This limitation arises from the threshold conditions, as illustrated in Figure 7, which reduce the
number of physically feasible equilibrium solutions. Specifically, because the Po di Maistra branch is substantially narrower
than the Po di Venezia, the model does not predict an avulsion event at this bifurcation. This outcome remains reasonable within



the model framework, as such a sudden reorganization of the delta would necessitate substantial width adaptation, which is beyond the current model's scope.

Nonetheless, comparison with field measurements of discharge partitioning under various flow regimes in the Po River Delta, as shown in Figure 8, indicates that the delta's present state aligns closely with a stable equilibrium where the primary channel, Po di Venezia, maintains dominance. Further comparison of model-derived bed profiles, established under formative conditions, with 2018 mean bed level data (Figure 9) reinforces these findings.

Historical bathymetric data indicate that mean bed levels within the delta branches have, in recent years, reverted to mid-20th
century values, following adjustments to accommodate anthropogenic impacts such as upstream dam construction, sediment extraction, and increased subsidence due to groundwater extraction. Evidence of a renovated equilibrium across many reaches of the Po River along its basin supports renewed sediment continuity, resulting in a greater sediment load reaching the delta. Given the aforementioned observations regarding the current equilibrium configuration of the delta, together with the channelization of the delta branches, sediment is transported efficiently to the delta front rather than contributing to natural subsidence
compensation within the delta plain. Recent satellite images analyses by Ninfo et al. (2018) further corroborate this trend, showing evidence of progradation along the main stem of the Po Delta.

As the main stem extends seaward, its hydraulic efficiency may diminish relative to the secondary branches, potentially increasing the likelihood of avulsion towards these channels (Salter et al., 2020; Barile et al., 2023).

The proposed model has also identified alternative equilibrium configurations without accounting for the lengthening of Po
di Venezia, which would enable possible avulsion at each bifurcation. Assuming no major changes to delta planform geometry, Figure 9 suggests that even the free-surface profile would be impacted by an upstream avulsion, particularly near the Po di Goro bifurcation. In such a scenario, the main channel's free-surface slope would alter, lowering the upstream elevation by approximately 1 meter, with significant implications for navigability and downstream infrastructure along the Po di Venezia. Most critically, a resulting backwater effect would influence upstream profiles, severely impacting bed equilibrium profile,
infrastructure stability, and levee integrity.

Historical records show that the Po di Goro has doubled its flow capture over the past century, now accounting for approximately 15% of the total upstream discharge, likely due to its shorter length relative to the main stem. Given the larger sediment supply along the main stem of the Po River, it is likely that progradation at Po di Venezia will keep at a faster rate than that occurring at the Po di Goro mouth, hence the probability of avulsion at this bifurcation is expected to increase substantially in
the coming decades.

## 6    Conclusions

This study advances the understanding of equilibrium dynamics in river-dominated deltas through a novel analytical model that underscores the role of internal feedback mechanisms within each deltaic bifurcation. The model effectively forecasts alternative equilibrium solutions and has identified likely avulsion sites in highly engineered deltas, with major implications
on variations in bed level and free surface elevation.



The application of the model to the Po River Delta has enabled the identification of a set of equilibrium configurations that best align with the measured flow discharge partitioning and the average bed elevation along the various delta branches. This exercise proved insightful, demonstrating that several alternative equilibrium configurations are possible, each characterized by flow discharge partitioning values significantly different from those currently observed. Furthermore, the study revealed that abrupt shifts from the present configuration to highly asymmetrical states in each bifurcation of the Po River Delta could result in substantial sediment deposition downstream, erosion upstream, and variations in free surface elevation. Such new equilibrium configurations, which would reduce flow discharge in the current main stem, are likely to have substantial consequences for downstream anthropogenic activities, including navigation, fisheries, and agriculture. Additionally, backwater effects could compromise the stability of upstream infrastructure.

These findings highlight the intricate interplay between fluvial and sedimentary processes, suggesting that future delta management strategies should consider internal feedbacks to enhance resilience against changing environmental pressures.

However, the assumption of cylindrical channels in the model presents limitations when compared to field data on bed levels. To better align with these data and refine insights on delta equilibrium, future work will adapt the model to incorporate variations in channel width.

In extending predictive capabilities, this research lays a foundation for more precise and sustainable approaches to delta conservation and management.

*Data availability.* Cross-section data are publicly available at the geoportale of the 'Agenzia Interregionale per il fiume Po' ($http : //geoportale.agenziapo.it$). The flow discharge partitioning measurements are published by the Veneto Regional Agency for Prevention and Protection of the Environment (ARPAV) in Zasso and Settin (2012).

**Appendix A: Derivation of the nodal point condition**

The transverse solid exchange is evaluated based on the procedure for describing two-dimensional bed load transport over an inclined bed (Ikeda et al., 1981):

$$q_{Ts}^* = q_{as}^* \left[ \frac{V^*}{\sqrt{U^{*2} + V^{*2}}} - \frac{r}{\sqrt{\vartheta_a}} \frac{\partial \eta^*}{\partial y^*} \right], \tag{A1}$$

where dimensional variables are denoted with the superscript $^*$. Specifically, $q_{Ts}^*$ represents the dimensional transverse solid discharge per unit width, and $q_{as}^*$ denotes the longitudinal solid flow discharge per unit width from upstream. $U^*$ and $V^*$ are the longitudinal and transverse velocity components, respectively. $\partial \eta^* / \partial y^*$ denotes the transverse bed slope at the junction of channels $b$ and $c$. The parameter $r$ is an experimental constant that ranges between $0.3$ and $1$ (Ikeda et al., 1981; Talmon et al., 1995). The Shields parameter, $\vartheta_a$, associated with the uniform flow is defined as:

$$\vartheta_a = \frac{Q_a^{*2}}{\frac{\rho_s - \rho}{\rho} g d_s^* C_a^2 D_a^{*2} W_a^{*2}} \tag{A2}$$



where $Q_a^*$ is the upstream flow discharge, $\rho_s$ and $\rho$ are the densities of sediment and water, respectively, g is the gravitational acceleration, $d_s^*$ is the sediment diameter, $C_a$ is the Chézy coefficient, $D_a^*$ is the flow depth, and $W_a^*$ is the channel width.

The velocity magnitude $\sqrt{U^{*2} + V^{*2}}$ is evaluated as the average flow velocity $U_{abc}^*$ within the cells length $\alpha W_a^*$, while the transverse velocity $V^*$ can be computed as:

$$V^* = \frac{Q_T^*}{\alpha W_a^* D_{abc}^*} \tag{A3}$$

Specifically, the average water depth at the bifurcation $D_{abc}^*$ is defined as:

$$D_{abc}^* = \frac{1}{2}\left(D_a^* + \frac{D_b^* + D_c^*}{2}\right) = D_{up}^* \frac{1}{2}\left(D_a + \frac{D_b + D_c}{2}\right) = D_{up}^* D_{abc}. \tag{A4}$$

In a similar fashion, $U_{abc}^*$ is defined as:

$$U_{abc}^* = \frac{1}{2}\left(U_a^* + \frac{U_b^* + U_c^*}{2}\right) = U_{up}^* \frac{1}{2}\left(U_a + \frac{U_b + U_c}{2}\right) = U_{up}^* U_{abc}, \tag{A5}$$

where $U_{up}^*$ can be rewritten as $\frac{Q_{up}^*}{D_{up}^* W_{up}^*}$. Noteworthy, each water depth $D_i$ and flow velocity $U_i$ are retrieved following the
scalings introduced in Eq. 6.

Furthermore, the transverse bed slope between the branches $\partial\eta^*/\partial y^*$ is defined as the bed level difference at the central point of the inlet of the downstream channels:

$$\frac{\partial\eta^*}{\partial y^*} = \frac{\eta_b^* - \eta_c^*}{\frac{W_b^*}{2} + \frac{W_c^*}{2}} = \frac{2D_{up}^*}{W_a^* r_a}(\eta_b - \eta_c) = \frac{1}{\beta r_a}(h_b - h_c - D_b + D_c), \tag{A6}$$

where the scaling $\eta_i = \frac{\eta_i^*}{D_{up}^*}$ has been applied and the bed elevation has been derived from $\eta_i = h_i - D_i$. Moreover, $r_a$ is the downstream enlargement parameter, and $\beta$ is the aspect ratio of the upstream channel defined as:

$$\beta = \frac{W_a^*}{2D_{up}^*}. \tag{A7}$$

Therefore, after a few manipulations and recalling the scaling in Eq. 6 the nodal point condition can be written:

$$q_{Ts} = \frac{q_T}{D_{abc}U_{abc}} - \frac{r}{\beta\sqrt{\vartheta_a}r_a}(h_b - h_c - D_b + D_c). \tag{A8}$$

*Author contributions.* LD, NT and MBP conceptualized the model. LD developed the model and wrote the first draft of the manuscript. NT and MBP reviewed the manuscript.

*Competing interests.* The authors declare no conflicts of interest relevant to this study.

*Acknowledgements.* This paper has been supported by the Italian Ministerial grant PRIN 2022 "Allogenic and Autogenic controls of DElta
MOrphodynamics (AADEMO)", n. 2022P9Z7NP—CUP D53D23004830006, by the Po River Basin Authority grant "Updating the Po



River Management Programme and integration with the delta branches"— CUP D33C22001030001, by the Italian Ministerial PRIN PNRR 2022: "Safety Equilibrium Conditions for rivers UndeR changing climatEs (SECURE)" n. P2022KA5CW— CUP D53D23022870001, and by the Italian Ministerial grant PRIN 2022 "Reconciling coastal flooding protection and morphological conservation of shallow coastal environments (Prot&Cons)", n. 2022FZNH82— CUP D53D23004660006.



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
