# Peer review of "Multiple Equilibrium Configurations in River-Dominated Deltas"

_EGUsphere, 2024_

## Author Comment (AC1)

Assessment: The manuscript is well-presented, relatively easy to read and follow, and is free of grammatical errors. Overall, I found the paper interesting and the results showing the variable effects of downstream length, in the channels immediately downstream or downstream of the second bifurcations, follows well-understood phenomena based on models from these authors and others. The comparison to the Po River delta is a welcomed piece, and the model does provide some interesting insights into the dynamics of that system. I have some criticisms of the contextualization of the results, especially with the assumption that flow is contained completely within channels. I am not suggesting that the authors redo anything, but it is my opinion that this assumption warrants some consideration and discussion. The abstract also needs to be edited to provide more substance and detail. At the moment it is very short and general to the point it is difficult to tell what novel insights this paper provides, and there are plenty of interesting results the authors could include to improve it.

Major points:

Abstract: The abstract lacks substantive information on the results of the paper. It I clear that equilibrium configurations are analyzed, but it's not clear what novel information or insights this manuscript provides. These authors have published several papers on equilibrium configurations in bifurcations, deltas, etc. and it would be beneficial to add specificity to the abstract.

Thank you for your valuable feedback. We agree that the abstract could be more effective in highlighting the key results of our study. We will revise it to provide greater specificity and clarity regarding the novel insights and contributions of our work.

It appears that there is no overbank flow allowed in such a model. If my understanding of the way these models work is correct, there cannot be an internal outlet. Is perfect flow conservation within the channel deltaic network to the seaward boundary realistic? For eg., see Allison et al (2023; 2012), Feizabadi et al., 2024; Gao et al., 2023; Hiatt & Passalacqua (2015;2017); Hiatt et al., (2018); Shaw et al., 2016. Even if it is not realistic, how does this assumption affect the results? Several studies suggest that connectivity with the floodplain is a primary driver of morphology in river deltas (e.g., Coffey and Shaw, 2017; Olliver and Edmonds 2021), and it stands to reason that this should affect bed morphology, etc. I recognize many of these examples are from Wax Lake Delta, but nevertheless this is often considered a prototype river dominated delta. I think this should be explained in the context of the assumption/limitations of the model design and in the discussion section to understand how containing flows to the channel network influences predictions of morphology/stability. This point may be important when considering the results shown in Figure 5. Water level asymmetry can drive lateral flow (Gao et al., (2023)) and may have implications for the results showing disagreement with the Po system, especially in the more downstream reaches (Tolle), where I assume the assumption of conserved channelized flow fails (I am not sure of this, of course, but there certainly looks to be connections in Figure 1b).

We sincerely appreciate the reviewer's insightful comments. The reviewer is absolutely correct in noting that the current model formulation does not include overbank flow. This was adopted as a first simplifying assumption to isolate the fundamental mechanisms governing the equilibrium of river deltas. Beyond the significant ecological implications, lateral flow dispersal primarily reduces flow velocity within channels, potentially enhancing local sediment deposition. However, as the reviewer rightly points out, this assumption may be overly restrictive when comparing the model to real-world deltas.

Nevertheless, the principle of flow conservation remains largely valid for many river deltas worldwide, such as the Po River Delta, where extensive human interventions, including widespread levee construction, have been implemented to confine the flow within designated channels. As noted by the reviewer, the degree of confinement decreases in the seaward reaches of the Tolle branch and the main channel, which likely contributes to the observed discrepancies in flow partitioning and bed elevation estimations in these areas.

Additionally, the model does not account for two secondary branches at the downstream end of the main channel. Field studies (Zasso & Settin, 2012) indicate that these branches (Busa di Tramontana and Busa di Scirocco) divert approximately 25% of the local flow discharge. This, combined with flow dispersal towards the lagoons at the delta fringes, introduces a degree of error in the local free-surface slopes and may influence the computed flow partitioning at the lowermost bifurcation considered.

Future studies could address these limitations by incorporating both localized and distributed flow losses into the model. However, implementing such modifications requires careful consideration when evaluating the long-term equilibrium of the system. A preliminary estimate of these contributions could be guided by site-specific field studies, though direct incorporation of such measurements assumes long-term constancy, which is highly uncertain, particularly in young and actively prograding deltas.

We will incorporate these considerations and the relevant contributions on perfect flow conservation within the channel and connectivity with the floodplain mentioned by the Reviewer into the discussion section of the manuscript, as we believe they provide an important context for interpreting the model's results. Additionally, we welcome further feedback on this matter and hope it fosters future collaborations.

Minor points and edits:

Line 10-11: Is the quantity and quality of sediment relative to the accommodation space really the key driver? In other words, even if the quantity of sediment is very small, if the space that needs to be filled is very small then a delta will be formed. The opposite is also true for large sediment loads with lots of space to fill. I suppose the point is moot because the authors mention this is in the next sentence, essentially.

Thank you for your comment. We will revise this introductory sentence accordingly to clarify the relationship between sediment supply and accommodation space.

Figure 1b: If the bathymetry of the channels is shown, we likely need a colorbar to distinguish elevations, otherwise it should be removed to match the birdsfoot. Also, I know the Po is the focus of the manuscript, but there should probably also be an inset map for the Birdsfoot. The inset map in Figure 1b is also not supremely helpful to those that are not familiar with northern Italy, so I'd recommend showing the full country and political boundaries.

Thank you for pointing this out. We will revise the figures accordingly, including improvements to the inset maps for both deltas. The colors in panel 1b were solely intended to highlight the course of the branches, as they would otherwise be difficult to distinguish. In the revised version of the

paper, we will only highlight the river axis of each branches with a single colour line in order to highlight the delta network that otherwise would be difficult to be captured.

Lines 21-22: "…drains a significant amount of water and carries a substantial quantity of sediment…" I'd recommend just reporting those annual figures here instead of using qualifying adjectives. Just give the quantities.

Thank you for the suggestion. We will add the appropriate values in the manuscript. Specifically, at the Po Delta Apex the mean annual discharge is approximately 1500 $m^3$/s and the corresponding total sediment load is in the range 9–12·$10^6$ tons/year (Lanzoni et al., 2015; Milliman & Farnsworth, 2013; Nienhuis et al., 2020).

Lines 30-32: This statement should be modified or removed: There are many studies focused on deltas that consider things other than flow routes and bio-ecology (not even sure what bio-ecology is). I would recommend a rewrite of this whole paragraph, giving proper consideration to the literature. Deltaic science is multidisciplinary are there are myriad studies across disciplines, so saying that most studies related to deltas "…focus merely on hydrodynamics…" is incorrect. The second argument also may not be correct – there are quite a number of detailed morphological modeling studies in the Mississippi River Delta, for example (e.g., Meselhe et al., (2021) cited below).

We acknowledge that our statement was overly broad and did not fully capture the extensive multidisciplinary research on deltaic systems. Our intention was to highlight the challenges in assessing the long-term equilibrium of river deltas. We will revise this paragraph to more accurately reflect the existing literature and provide a more comprehensive discussion of relevant studies.

Lines 36-42: While I am familiar with these models and agree that they are useful , I recommend being more objective and removing words such as "easier" and "powerful insight."

Thank you for your feedback. We will revise these sentences to ensure a more objective tone.

Line 57-58: There is some work on the bed morphology in Wax Lake Delta from Ehab Meslhe's group using a Delt3D Morpho model (Meselhe et al., 2021).

In this context, we are specifically referring to morphodynamic models applied to the Po River Delta. However, we acknowledge the relevance of the work by Meselhe et al. (2021) and will consider referencing it where appropriate.

Line 182: What is the critical threshold for R_Up? Can the authors please present an example for on of the L'tot values so the reader can more easily contextualize the results in Figure 4?

Here the R_cr is considered as the point where the single solution at DQ=0 splits in the 3 separate solutions. In other words when each single line in Figure 4 bifurcates moving leftward. To improve clarity, we will add a filled dot, matching the color of the corresponding lines, in Figures 4 and 5 to indicate this threshold.
Regarding L'tot, it is defined in eq. (14) of the manuscript. It basically represents the distance of the bifurcation from the sea, aggregating the length of the downstream branches.

To get a first estimate of the value of R_cr, one can use the formulation for the single bifurcation as defined in Durante et al. (2024). Manipulating equation 28 of such paper we retrieve:

$$R_{cr} = \frac{3}{4} \frac{\left(^7/_3 L - ^3/_2 Fr^2\right)}{\left(6L + 3Fr^2 + 4\,L\,Fr^2\right)}$$

Where, in order to compare with L'tot, L need to be formulated as:

$$L = \frac{L'_{tot}\,W^*_{up}\,s_{up}}{D^*_{up}}$$

with W, D and s are the width, depth and slope of the upstream channel.

Citations

Allison, M. A., Meselhe, E. A., Kleiss, B. A., & Duffy, S. M. (2023). Impact of water loss on sustainability of the Mississippi River channel in its Deltaic Reach. Hydrological Processes, 37(10), e15004.

Allison, M. A., Demas, C. R., Ebersole, B. A., Kleiss, B. A., Little, C. D., Meselhe, E. A., ... & Vosburg, B. M. (2012). A water and sediment budget for the lower Mississippi–Atchafalaya River in flood years 2008–2010: Implications for sediment discharge to the oceans and coastal restoration in Louisiana. Journal of Hydrology, 432, 84-97.

Coffey, T. S., & Shaw, J. B. (2017). Congruent bifurcation angles in river delta and tributary channel networks. Geophysical research letters, 44(22), 11-427.

Gao, W., Wang, Z. B., Kleinhans, M. G., Miao, C., Cui, B., & Shao, D. (2023). Floodplain connecting channels as critical paths for hydrological connectivity of deltaic river networks. Water Resources Research, 59(4), e2022WR033714.

Feizabadi, S., C. Li, and M. Hiatt (2024), Response of river delta hydrological connectivity to changes in river discharge and atmospheric frontal passage, Frontiers in Marine Science, 11, https://doi.org/10.3389/fmars.2024.1387180

Hiatt, M. and P. Passalacqua (2015), Hydrological connectivity in river deltas: The first-order importance of channel-island exchange, Water Resources Research, 51, 2264–2282, https://doi.org/10.1002/2014WR016149

Hiatt, M. and P. Passalacqua (2017), What controls the transition from confined to unconfined flow? Analysis of hydraulics in a coastal river delta, Journal Hydraulic Engineering, 60th Anniversary Reviews,143(6), https://doi.org/10.1061/(ASCE)HY.1943-7900.0001309

Hiatt, M., E. Castañeda-Moya, R. Twilley, B.R. Hodges, and P. Passalacqua (2018), Channel-island connectivity affects exposure time distributions in a coastal river delta, Water Resources Research, 54, https://doi.org/10.1002/2017WR021289

Meselhe, E., Sadid, K., & Khadka, A. (2021). Sediment distribution, retention and morphodynamic

analysis of a river-dominated deltaic system. Water, 13(10), 1341.

Olliver, E. A., & Edmonds, D. A. (2021). Hydrological connectivity controls magnitude and distribution of sediment deposition within the deltaic islands of Wax Lake Delta, LA, USA. Journal of Geophysical Research: Earth Surface, 126(9), e2021JF006136

Shaw, J. B., Mohrig, D., & Wagner, R. W. (2016a). Flow patterns and morphology of a prograding river delta. Journal of Geophysical Research: Earth Surface, 121, 372–391. https://doi.org/10.1002/2015JF003570

Reference

Durante, L., Bolla Pittaluga, M., Porcile, G., & Tambroni, N. (2024). Downstream control on the stability of river bifurcations. *Journal of Geophysical Research: Earth Surface*, *129*(10), e2023JF007548.

Lanzoni, S., Luchi, R., & Pittaluga, M. B. (2015). Modeling the morphodynamic equilibrium of an intermediate reach of the Po River (Italy). Advances in Water Resources, 81, 95-102.

Meselhe, E., Sadid, K., & Khadka, A. (2021). Sediment distribution, retention and morphodynamic analysis of a river-dominated deltaic system. Water, 13(10), 1341.

Milliman, J. D., & Farnsworth, K. L. (2013). River discharge to the coastal ocean: a global synthesis. Cambridge University Press.

Nienhuis, J. H., Ashton, A. D., Edmonds, D. A., Hoitink, A. J. F., Kettner, A. J., Rowland, J. C., & Törnqvist, T. E. (2020). Global-scale human impact on delta morphology has led to net land area gain. Nature, 577(7791), 514-518.

Zasso, M. and Settin, T.: Sulla ripartizione delle portate del Po tra i vari rami e le bocche a mare del delta: esperienze storiche e nuove indagini all'anno 2011, Veneto Regional Agency for Prevention and Protection of the Environment (ARPAV), Relazione 02/12, https://www.arpa.veneto.it/temi-ambientali/idrologia/file-e-allegati/idrologia-del-delta-del-po/510 ripartizione-delle-portate-nel-delta-del-po---esperienze-storiche-e-nuove-indagini-al-2011.pdf, 2012.

---

## Author Comment (AC2)

This research article presents a numerical framework that schematizes a delta channel network as a series of connected bifurcations. The research builds on a large body of work that considers "stability of bifurcations", or what configuration of water and sediment partitioning at a bifurcation enables the bifurcation to persist (i.e., be in equilibrium) rather than abandon one of the branches. This work extends that framework to multiple bifurcations, wherein the upstream leg of a bifurcation is treated as one of two downstream branches of another bifurcation; this is considered to be analogous to a delta network. Through a system of equations derived for any initial delta configuration, system parts and variables are isolated, and their stability is determined. A key finding, per the article title, is that there are multiple flow-partitioning configurations of given planform delta configuration that are stable. The framework is then initialized with data from the real-world Po River delta system, and the stability of this system is explored. Through this analysis, the authors have identified new insights into the possible controls on avulsion and bifurcations stability, as well as potential futures for the Po River delta system.

Overall, the article is fairly well written, interesting, and will be well received by the readers of Earth Surface Dynamics. The introduction and discussion could benefit from clarification to contextualize the research. The model description and presentation of results are excellent. In particular, I enjoyed reading the Po River delta application, and exploration of possible avulsions in the system (Figure 9). I recommend some minor revisions before publication.

Main comments:

1. The motivation in the Introduction could be made more specific to this research. At present, it is very general about anthropogenic modification and "proper management" but does not specifically talk about channels or avulsions. For example, there is discussion of levees reducing flow onto interdistributary basins (line 27) but it is not clear how this relates to channel stability. The authors mention "navigability and downstream infrastructure" (line 388) in the discussion, which may be more relevant motivations for this study.

Thank you for this insightful comment. We agree about the introduction to be more specific to the focus of our study. We will revise this section to better highlight the relevance of channel stability and possible avulsion sites, explicitly mentioning their impact on navigability and infrastructures.

2. I am not incredibly familiar with the Salter 2018 2020, Barile 2023, Ragno 2022, and Durante 2024 papers and the framework described in each (lines 36–54). It would help the reader understand the advance of this study if it could be clarified how each of these models/stability frameworks differs (or not) from the one presented here. This will, overall, help to contextualize this study in the wider literature.

Thank you for your comment. We will rephrase the paragraph to clearly distinguish which models correspond to the single bifurcation and which pertain to the entire delta, thereby better highlighting how the frameworks in the cited studies differ from or align with the one presented in our work.

3. a. The organization of the discussion could be improved. The Discussion could be improved by reducing the number of paragraphs and grouping logically-related ideas into subsections. Subsections could break apart the analysis of (i) internal bifurcation feedback, (ii) system planimetric effects, (iii) and Po River delta application.

Thank you for this constructive suggestion. We agree that the organization of the Discussion section could be improved for clarity and coherence. We will introduce distinct subsections as suggested to enhance readability and better highlight the key aspects of our analysis.

b. I also suggest the authors revisit the logical organization of their paragraphs in the Discussion section. For example, there are several times that the discussion mentions seaward effects (lines 382, 393), but these ideas span a few (sometimes short) paragraphs.

Thank you for your suggestion. We will review the organization of the Discussion section to ensure that related ideas are presented in a more cohesive manner.

4. The idea of adjustment timescales and equilibria introduced in lines 307–312 is not revisited when discussing the channel abandonment (line 344–348), or soft avulsion (line 354–360), or delta lobe progradation (line 382–390). In my opinion, this discussion of timescales is the most important aspect of applying this numerical framework to the real world in any meaningful way, which seems to be of interest to the authors. I realize the framework is not fully morphodyanmic and does not explicitly include a temporal evolution, but the authors could identify terms in the framework that would be compared against real world processes and rates mentioned above (abandonment, soft avulsion, lobe progradation) to determine the scales at which this framework is useful. To me, this is a major limitation to understanding whether this framework has any predictive power.

Thank you for your insightful comment. We acknowledge the importance of discussing timescales in relation to equilibria and the key phenomena characteristic of river deltas. The reviewer is correct in noting that the current model formulation does not incorporate temporal evolution, instead relying on the concept of equilibrium. However, we can provide a rough estimate of the morphological timescale $T_M$, which gives an order of magnitude for the temporal scale over which the system evolves. This timescale is defined as:

$$T_M = (1 - p) \frac{W_{up}^* D_{up}^*}{q_{S_{up}}^*}$$

where W, D and qs represent the width, depth, and solid discharge at the delta apex, respectively. However, we recognize that real-world deltas may respond differently due to varying external factors (e.g., sea-level rise, subsidence, bank erosion, vegetation), and the temporal scales at which these changes occur. Nonetheless, equilibrium-based models remain valuable for identifying dominant processes and feedback mechanisms within the system, as discussed in Zhou et al. (2017). We will revise the manuscript to better highlight these points and clarify the applicability of the model in real-world scenarios.

Zhou, Z., Coco, G., Townend, I., Olabarrieta, M., Van Der Wegen, M., Gong, Z., ... & Zhang, C. (2017). Is "morphodynamic equilibrium" an oxymoron?. *Earth-Science Reviews*, *165*, 257-267.

Minor comments/corrections:

- The meaning of "multiple equilibrium states" on line 4 of the abstract is not clear at this point. I suggest revising the abstract to be more specific about the "unique challenges" facing deltas (see Main Comment 1) and more specific about the numerical approach before stating what exactly the study identifies.

  We agree. We will revise the abstract.

- The actual description of the numerical framework, including relevant terms and their relationships, is excellent. Thank you.

  Thank you.

- The analysis beginning on line 168 assumes a symmetrical planform delta (Lb1=Lc1), correct? This was not clear to me at first: even though it does say symmetrical on line 171/172, the sketch in Figure 3 is not depicting a symmetrical delta, but this sketch is referenced on line 172. Moreover, what does Figure 3 depict that is not already covered in Figure 2? I found this to be a sort of confusing point, because I couldn't understand how Lb1=Lb2 in Figure 3 when they are clearly different, until I reread a few times and realized the sketch did not match the description. I suggest the authors consider revising Figures 2/3/4c/4d to show the necessary components and only one time, for both a case of symmetric delta asymmetric delta. This will also help clarify how a delta can be planform symmetrical but have asymmetrical discharge partitioning.

  The text introducing Figure 3 has been revised to clarify the original source of confusion.

- Suggest to indicate the meaning of the dashed lines in Figures 6 and 7 in either the figure itself or the figure caption.

  Thank you for the suggestion. We will add a sentence to the figure caption to explicitly clarify the meaning of the dashed lines in Figures 6 and 7.

- "the concept of long-term morphodynamic equilibrium in river deltas may be inherently transient" was a confusing statement to me. I don't think the authors mean the concept is transient. Suggest revising to be more specific.

  Thank you, we will rewrite the sentence.